# Polyphenolic Compounds Nanostructurated with Gold Nanoparticles Enhance Wound Repair

**DOI:** 10.3390/ijms242417138

**Published:** 2023-12-05

**Authors:** Adriana Martínez-Cuazitl, María del Consuelo Gómez-García, Salvador Pérez-Mora, Marlon Rojas-López, Raúl Jacobo Delgado-Macuil, Juan Ocampo-López, Gustavo Jesús Vázquez-Zapién, Mónica Maribel Mata-Miranda, David Guillermo Pérez-Ishiwara

**Affiliations:** 1Laboratorio de Biomedicina Molecular, ENMyH, Instituto Politécnico Nacional, Mexico City 07320, Mexico; admartinezc@ipn.mx (A.M.-C.); consuelogg22@yahoo.com.mx (M.d.C.G.-G.); sperezm1510@alumno.ipn.mx (S.P.-M.); 2Escuela Militar de Medicina, Centro Militar de Ciencias de la Salud, Universidad del Ejército y Fuerza Aérea-Secretaría de la Defensa Nacional, Mexico City 11200, Mexico; mmcmaribel@gmail.com; 3Centro de Investigación en Biotecnología Aplicada, Instituto Politécnico Nacional, Santa Inés Tecuexcomac 90700, Mexico; marlonrl1@hotmail.com (M.R.-L.); rdmacuil@yahoo.com.mx (R.J.D.-M.); 4Laboratorio de Histología e Histopatología del Área Académica de Medicina Veterinaria y Zootecnia, Universidad Autónoma del Estado de Hidalgo, Tulancingo de Bravo 42090, Mexico; jocampo@uaeh.edu.mx; 5Centro de Investigación y Desarrollo del Ejército y Fuerza Aérea Mexicanos de la Secretaría de la Defensa Nacional (CIDEFAM—SEDENA), Mexico City 11400, Mexico; gus1202@hotmail.com

**Keywords:** polyphenolic-gold nanoparticles conjugates, wound healing model, FTIR analyses of wound repair tissue

## Abstract

Gold nanoparticles (AuNPs) have been used in a wide range of applications, conferring to bio-molecules diverse properties such as delivery, stabilization, and reduction of the adverse effects of drugs or plant extracts. Polyphenolic compounds from *Bacopa procumbens* (*B. procumbens)* (BP) can modulate proliferation, adhesion, migration, and cell differentiation, reducing the artificial scratch area in fibroblast cultures and promoting wound healing in an in vivo model. Here, chemically synthesized AuNPs conjugated with BP (AuNP-BP) were characterized using UV-Vis, ATR-FTIR, DLS, zeta-potential, and TEM analysis. The results showed an overlap of the FTIR spectra of the polyphenolic compounds from *B. procumbens* adhered to the surface of the AuNPs. UV-vis analysis indicated that the average size of the AuNP-BP was 28 nm, while DLS analysis showed a size of 44.58 nm and, by TEM, a size of 16.5 nm with an icosahedral morphology was observed. These measurements suggest an increase in the size of the nanoparticles after conjugation with BP, compared to the sizes of 9 nm, 44.51 nm, and 14.17 nm for the unconjugated AuNPs, respectively. Furthermore, the zeta potential of the AuNPs, which was originally −36.3 ± 12.3 mV shifted to −18.2 ± 7.02 mV after conjugation with BP, indicating improved stability of the nanoparticles. Enhancement of the wound healing effect was evaluated by morphometric, histochemical, and FTIR changes in a rat wound excision model. Results showed that the nanoconjugation process reduced the BP concentrations by 100-fold to have the same wound healing effect as BP alone. Besides, histological and FTIR spectroscopy analyses demonstrated that AuNP-BP treatment exhibited better macroscopical performance, showing a reduction in inflammatory cells and an increased synthesis and improved organization of collagen fibers.

## 1. Introduction

Wound healing is a complex and coordinated process that involves different kinds of cells and molecular signaling. The main objective of this process is to restore physiological and esthetic properties. The last step involves the substitution of collagen type III by collagen type I and the reordering of both types of fibers to improve the mechanical properties. Injuries remain a clinical problem due to early and late complications. Meanwhile, the incidence of chronic wounds has rapidly increased due to the rising prevalence of type 2 diabetes, peripheral vascular disease, and metabolic syndrome [1,2,3].

Even though fast and optimal wound closure are the main goals of wound care, no topically effective medication has yet been developed to accelerate the wound healing process or to prevent abnormal wound healing [4].

For millennia, humans have been treating their wounds with traditional wound management, using medicines derived from local plants, animals, and natural products. Since traditional medicine is empirically used, the active ingredients in it have been studied and must be validated by experimental controlled assays [5].

Active components of medicinal plants have been studied for their variety and effectiveness in the wound healing process [6]. The wound healing effects of phenolic compounds, including flavonoids, secoiridoids, phenolic acids, phenolic alcohols, and lignans, have been studied due to their antioxidant, antimicrobial, and/or bio-stimulatory properties that are related to tissue regeneration [7]. *Bacopa procumbens* (*B. procumbens*), a member of the Scrophulariaceae family, has been used in traditional medicine for skin wound healing. Recently, our group demonstrated that its polyphenolic compounds (BP) regulated proliferation, cell adhesion, and enhanced fibroblast migration in in vitro assays. On the other hand, in the rat model, BP accelerated the wound healing process by at least 48 h, reducing inflammation, increasing cell proliferation, and promoting the deposition and organization of collagen by regulating both the canonical and non-canonical TGF-β1 expression pathways [8]. Plant compounds derived from traditional medicine are often preferred due to their low cost and limited adverse effects. However, to improve their bioavailability and efficacy, different nanomaterials have been used to nanostructure these compounds [6].

Nanoformulation has been successful in transporting, protecting, and delivering active drugs, aiming to obtain synergistic activity and/or enhance the response at the bio-interface [4,9]. Particularly, due to their unique physicochemical properties and biocompatibility, gold nanoparticles (AuNPs+) have been used in a broad range of applications, including genomics, biosensors, immunoanalysis, clinical chemistry, diagnosis, and pharmaceutics. This usage confers upon the conjugated biomolecules diverse properties such as delivery, stabilization, and reduction of adverse effects of drugs or plant extracts [10].

The effect of AuNPs in enhancing the activity of some compounds has been assessed in skin tissue repair [11,12]. However, their ability to enhance the pharmacological activity of conjugated drugs depends on the shape and size of the synthesized nanoparticles. Different methods for nanostructuring AuNPs have been used; the most common method is the reduction of Au (III) using citrate, as proposed by Turkevich [13]. 

To characterize the nanostructures generated, techniques such as the physical Surface Plasmon Resonance (SPR) phenomenon, Transmission Electron Microscopy (TEM), Scanning Electron Microscopy (SEM), Energy Dispersive X-ray Spectroscopy (EDX), Atomic Force Microscopy analysis (AFM), X-ray Diffraction (XRD), UV-Visible (UV-Vis) and Fourier Transform Infrared (FTIR) spectroscopies, Raman spectroscopy, and Electrophoretic Light Scattering (ELS) can be used [14].

Here, to determine the usefulness of AuNPs in enhancing the effect of the polyphenolic compounds from *B. procumbens*, a nanoformulation (AuNPs-BP) was synthesized and characterized by different physicochemical techniques and tested topically in the rat excision wound model through morphometric, histological, and chemometric analyses.

## 2. Results

### 2.1. UV-Visible Spectrophotometry Analyses of Nanofunctionalized Gold Nanoparticles with Polyphenolic Compounds of B. procumbens (AuNP-BP)

The synthesis of gold nanoparticles (AuNPs) was performed using the modified Turkevich method, which resulted in the red color of the obtained colloid. Different concentrations of polyphenolic compounds from *B. procumbens* were conjugated as described in the methodology. The conjugation process changed the red color of the AuNPs to brown; the intensity of the brown color increased proportionally with the BP concentration. To determine the adequate concentration of the extract that covers the surface of the AuNPs, the conjugations (AuNP-BP) were characterized by UV-visible spectrophotometry. The UV-visible absorption spectra of AuNPs and AuNP-BP using several concentrations of BP (0.4, 0.8, 1.6, 3.2, and 6.4 mg/mL) are shown in the Figure 1. Single AuNPs showed an intense band centered at 520 nm which is associated with the surface plasmon resonance (SPR). The SPR band of AuNP-BP presented a 4 nm small spectral shift to low energies with respect to AuNPs. At higher concentrations of BP, the AuNP-BP showed an intense band at 668 nm, which arises from chlorophyll α [15]. However, the conjugate at 1.6 mg/mL of BP concentration showed the higher intensity of the SPR band without the chlorophyll α peak at 668 nm. Thus, this concentration of BP in the conjugate AuNP-BP was chosen to be used in the in vivo healing model; the shape was 28 nm at 2.89846 × 10^11^ AuNP-BP/mL. The concentration of nanoparticles estimated by the Haiss method through the UV-vis spectrum was 3.83 × 10^13^ particles/m. So, each particle contained 4.18 ag/p of BP extract on its surface.

#### 2.1.1. FTIR Spectroscopy of the AuNP-BP

AuNPs, BP, and conjugate AuNP-BP at 1.6 mg/mL were analyzed by Fourier transform infrared (FTIR) spectroscopy in the fingerprint region (1800–900 cm^−1^) (Figure 2). A high similarity between the FTIR spectra from both BP and AuNP-BP at 1.6 mg/mL was observed. Such similarity is attributable to the fact that the AuNPs were conjugated with BP molecules from the extract, whereas the AuNPs alone only have citrate groups on their surface, which were formed during the synthesis process. 

Table 1 shows band frequencies and functional groups corresponding to the FTIR spectra of AuNPs, BP extract, and AuNP-BP at 1.6 mg/mL. The bands at 1710 cm^−1^, 1637 cm^−1^, 1589 cm^−1^, 1470 cm^−1^, and 1402 cm^−1^ observed in AuNPs are related to N-H and C-O, N-H, CH_2_-O, C-C, and C=O vibrations from the citrate functional group, respectively [16,17]. In the FTIR spectra of both AuNP-BP and BP, several bands directly related to the BP chemical compounds were observed. The band at 1700 is related to the C=O ester bond; at 1610 is the aromatic double bond; the band at 1512 cm^−1^ is related to C-O aromatic bond of phenolic compound; the 1450 band is related to aromatic C-C bond; band of 1375 is related to O-H bending [18], 1258 cm^−1^ and 1230 cm^−1^ arise for C-O on polyols, and 1171 cm^−1^, 1076 cm^−1^, 1045 cm^−1^ arise from C-O from carbohydrates or primary, secondary, and tertiary alcohols, respectively, present in the BP extract [19].

**Table 1 ijms-24-17138-t001:** FTIR band frequencies and the related functional groups observed in gold nanoparticles (AuNPs), BP, and AuNP-BP conjugate.

Band	AuNPs	BP	AuNP-BP 1.6 mg/mL	Functional Group	References
1	1742	-	-	O-H on AuNPs	[16]
2	1712	-	-	N-H on AuNPs	[16]
3	-	1703	1703	C=O	[18]
4	1637	-	-	N-H on AuNPs	[17]
5	1612	1608	1608	aromatic double bond	[18]
6	1589	1589	1589	CH_2_-O on AuNPs	[17]
7	1512	1512	1510	C-C flavonoids and aromatic rings	[19]
8	1470	-	-	C-C on AuNPs	[16]
9	1402	-	-	C=O on AuNPs	[17]
10	-	1258	1259	C-O on polyols	[19]
11	-	1230	1229	C-O on polyols	[19]
12		1171	1165	C-O and -OH of primary alcohols	[19]
13	1076	1076	1074	C-O alcohols, phenols, carboxylic anions	[16,19]
14	-	1045	1055	C-O and -OH of tertiary alcohols	[19]

#### 2.1.2. Dynamic Light Scattering Measurements (DLS) and Electrophoretic Light Scattering (ELS)

Particle size in their dispersed state in water and the stability of the colloidal system, converted to zeta-potential, were determined. DLS measurement showed that the size of AuNPs was 44.51 nm, while that of the AuNP-BP was 44.58 nm. The zeta-potential of AuNPs was −36.3 ± 12.3 mV; after conjugation with the BP (AuNP-BP), the zeta-potential was reduced to −18.2 ± 7.02 mV (Figure 3).

#### 2.1.3. Transmission Electron Microscopy (TEM) of AuNP and AuNP-BP

AuNP and AuNP-BP colloidal dispersions were analyzed by TEM, where their icosahedral morphology was observed, as shown in Figure 4 a,b. Additionally, Figure 4 c,d show the size distributions corresponding to AuNP and AuNP-BP. The average particle size, obtained by image processing with ImageJ software, was 14.17 ± 0.05 nm for AuNP, and 16.5 nm for AuNP-BP.

Furthermore, the AuNP samples exhibit minimal interaction or proximity between particles. In contrast, the AuNP-BP samples show enhanced interaction, leading to a more pronounced aggregation of the particles.

### 2.2. Effects of Topical Application of AuNP-BP Hydrogel in a Rat Wound Healing Model

#### 2.2.1. Macroscopic Changes Induced by AuNP-BP Hydrogel

Macroscopic analysis of wound healing showed improved wound reduction using topical hydrogel with BP or with AuNP-BP hydrogel, in comparison to the control group without treatment, or to the group treated only with AuNPs. On day 0, the size of the wounds in all groups was statistically similar as showed on representative images on Figure 5a. On day 7, wounds treated with BP or AuNP-BP hydrogels presented a moderate scab, while in the group without treatment (WOT) or in the animals treated with AuNPs alone, the scab was more prominent (Figure 5a). At this time, the reduction of the wound area was significantly greater in the BP and AuNP-BP hydrogel groups, reducing it by approximately 85%, compared to the WOT and AuNP groups, in which the wound area was reduced only by 67% and 73%, respectively (Figure 5b).

#### 2.2.2. AuNP-BP Hydrogel Enhances Histological Wound Healing

Histopathological analyses of healing on day 7 of the injured animal groups, whether without treatment or treated with AuNPs, revealed an important inflammatory infiltration, predominantly of neutrophil cells, macrophages, and a few fibroblasts. There were few blood vessels close to the epidermal region that were evident, and non-re-epithelialized tissue was found. Instead, healings from groups treated with BP or with AuNP-BP hydrogels displayed less inflammation and a higher number of fibroblasts. Interestingly, the injuries presented more blood vessels in the dermis, and an important re-epithelization process was evident, although it was still thin, with few strata (Figure 6 and Figure 7). Masson’s trichrome staining of the WOT and AuNP groups revealed some disorganized fibroblasts and an incipient formation of collagen fibers, along with an important inflammatory cell population. However, the PB and AuNP-BP groups displayed a significant quantity of elongated fibroblasts and a better distribution and organization of collagen fibers (Figure 6 and Figure 7).

Figure 8 shows that wounds from the WOT group presented slight, thin, and disorganized green fibers (collagen type III), while wounds treated with BP and AuNP-BP hydrogels showed thick and organized red-yellow fibers (collagen type I).

#### 2.2.3. ATR-FTIR Spectra Changes on Skin Wound Healing Induced by AuNP-BP

Figure 9 displays the average spectra of skin wound healing for the WOT, AuNPs, BP, and AuNP-BP groups at 7 days post-treatment. The main biomolecular characteristics of the biological samples are evident, including lipids, proteins, carbohydrates, and nucleic acids (Figure 9 and Table 2). The average spectra of treated skin wound healing showed distinct collagen bands, with two intense bands at 1660 cm^−1^ and 1549 cm^−1^, which are related to the stretching vibration of C=O functional groups in collagen. There was also a combination of C-N stretching and N-H bending vibration in the triple helix of collagens at 1338 cm^−1^ (CH_2_ side chain vibrations), 1286 cm^−1^, and 1204 cm^−1^, which are related to CH_2_ wagging vibration from the glycine backbone and proline side chain. Additionally, there was a peak at 1082 cm^−1^, assigned to C-O stretching vibrations of the carbohydrate residue [20,21]. The bands at 1737 cm^−1^ (C=O) and 1456 cm^−1^ (CH_2_) are related to lipids, whereas the band at 1400 cm^−1^ arises from CH3 vibrations of GAGs (glycosaminoglycans). Furthermore, phospholipid groups are associated with absorption peaks at 1268 cm^−1^ and 1085 cm^−1^ [20,21,22].

The major peaks with significant shifts are related to Amide I, Amide II, CH_2_ lipids, PO_2_^−^ *_asym_* Phospholipds, and PO_2_^−^ *_sym_* Phospholipds (Figure 10 and Table 3). 

Highlight refer to statistical significance in each group.

The absorbances related to Amide II (1549 cm^−1^), CH_2_ lipids, and collagen type I (1456 cm^−1^) as well as CH_2_ collagen (1286 cm^−1^), decreased in the AuNP-BP group in comparison with WOT or with AuNPs. Meanwhile, the absorbance of CH_3_ GAGs (1400 cm^−1^) increased between BP and AuNP-BP versus AuNPs. Bands related to CH_2_ collagen (1204 cm^−1^), C=O carbohydrates (1161 cm^−1^), and PO_2_^−^ *_sym_*(1085 cm^−1^) were increased on BP and AuNP-BP in comparison with AuNPs (Figure 11 and Table 4) [20,21,22].

Highlight refer to statistical significance in each group.

Results in Figure 12a represent the average of the second derivative absorbance related to Amide I, showing the band related to β-sheet at 1665 cm^−1^ and the band related to α-helix at 1660 cm ^−1^ [22]. We found a significant increase of α-helix/β-sheet on treatment with AuNP-BP hydrogel in comparison to the other groups of treatments (WOT, AuNPs, and BP), as can be seen in Figure 12b. 

## 3. Discussion

Metallic nanoparticles have gained much attention as powerful materials for nanomedicine, catalysis, environmental research, biosensors, and drug delivery due to their physicochemical properties [23]. 

Since AuNPs are synthesized by chemical reduction with sodium citrate, at the end of the synthesis, they are covered on their surface by citrate ions, which confer a negative charge to the particles. The electronegativity of the gold nanoparticles generates an electrostatic attraction to the multiple antioxidant compounds present in the *Bacopa procumbens* (*B. procumbens*) extract during the conjugation process. This explains why the FTIR spectrum of the BP-conjugate is observed to be very similar to the spectrum of the BP extract. On the one hand, gold nanoparticles exhibit the surface plasmon resonance phenomenon, which is responsible for their optical properties, allowing their size, shape, and composition to be tracked and easily controlled [17]. On the other hand, surface modification of AuNPs provides a stable matrix for the biomolecular functionalization of several molecules, such as proteins, organic compounds, or plant extracts, among others. Plant extracts, such as BP, are used as capping agents due to their phytochemical composition, enriched with flavonoids, phenols, and terpenoids [8,23]. 

In this study, we conjugated the AuNPs with the bioactive polyphenolic compounds (BP) of *B. procumbens*, which promote tissue repair and wound healing [8]. The characterization of AuNPs synthesis and AuNP-BP conjugation was performed using UV-visible spectrophotometry, FTIR spectra, DLS, TEM and electrophoretic light scattering to determine their optical properties, size, concentration, agglomeration state, hints on NP shape, surface composition, ligand binding, and hydrodynamic size [14,24]. The conjugation of BP to AuNPs shifted the nanoparticles’ absorbance from 520 nm to 524 nm. The resonance wavelength and bandwidth of AuNPs depend on the particle size and shape [17]. 

The Lambert-Beer law indicates that absorbance is directly proportional to the extinction coefficient multiplied by the path length and the concentration of the solution. Conversely, Mie and coworkers demonstrated that the oscillation modes depend on the particle size and as the size decreases, the maximum absorption also decreases, events that we observed when AuNPs were conjugated with BP [24,25].

Higher concentrations of AuNP-BP displayed a peak at 668 nm, related to the chlorophyll α. Some reports have shown that higher concentrations of extracts could act as capping and stabilizing agents, such as phenolics, flavonoids, alkaloids, polysaccharides, saponins, tannins, and organic acids, leading to Au-hydro-complex formation rather than nanoparticles [24,25]. Thus, we chose a BP concentration of 1.6 mg/mL for optimal capping formation, while the concentration of nanoparticles, estimated by the Haiss method through the UV-vis spectrum, was 3.83 × 10^13^ particles/mL. Therefore, each particle contained 4.18 ag/p of BP extract on its surface [15,23].

According to the FTIIR analyses of BP and AuNPs-BP, peaks at 1700 cm^−1^, 1610 cm^−1^, 1512 cm^−1^, 1450 cm^−1^, 1375 cm^−1^, 1258 cm^−1^, 1230 cm^−1^, 1171 cm^−1^, 1076 cm^−1^, and 1045 cm^−1^ suggested that the AuNPs were capped with BP polyphenols, as previously reported [8]. This is similar to other reports where polyphenols derived from alcoholic extracts of natural resources have been used to cap gold nanoparticles [18,19].

The size of nanoparticles can vary according to the instrumentation method; it can change due to various interaction forces in the solution, including Van der Waals forces. Polyphenols limit particle growth and prevent agglomeration, thereby stabilizing the gold nanoparticles [23]. In this research, we found that gold nanoparticles alone measured around 9 nm by UV-Vis spectra and changed to 28 nm when capped with BP. When measured by DLS, the initial size of AuNP was 44.51 nm, increasing to 44.58 nm after BP conjugation. However, the average particle size obtained by TEM image processing with ImageJ software was 14.18 nm for AuNP, and 16.5 nm for AuNP-BP. In concordance with our results, other authors have also reported an increase in size when capping is performed [23]. The increment in AuNP-BP sizes measured by the three methods suggests that the *B. procumbens* compounds, mainly polyphenols, envelop the core of the AuNPs, thereby stabilizing the gold nanoparticles.

The zeta potential provides valuable information about the surface charge as well as the stability of AuNP-BP in colloidal systems [26]. The AuNP-BP potential was −18.2 ± 7.02 mV, indicating significant stabilization; the conjugation of BP compounds increased the AuNPs’ stability, conferring a higher negative zeta potential. This modification in charge can influence several properties of the nanoparticles, including their stability in suspension, their interaction with biological cells or tissues, and their potential as a vehicle for drug delivery. In addition, it is known that a highly negative zeta potential value indicates that there is enough repulsive force to prevent the aggregation of gold particles, with values lower than −30 mV considered strongly anionic. BP, through the functional groups of polyphenols (-OH and -COOH), provides a negative charge, as reported in other studies [23].

Interestingly, using a hydrogel containing AuNP-BP at 1.6 mg/mL BP concentration for wound treatment in a wound excision animal model, we showed an increased percentage of wound reduction seven days after treatment, in contrast to animals treated with hydrogel containing AuNPs alone, suggesting that the capping compounds derived from BP induced these wound healing effects, as we previously described with BP alone [8]. 

Our findings are also in concordance with other reports that showed that gold nanoparticles capping with natural products, such as quercetin derived from *Abelmoschus esculentus* (L.) (okra), or with epigallocatechin gallate and α-lipoic acid on linear wound enhanced wound healing [6,27,28].

BP and AuNP-BP compounds not only showed better macroscopic effects, they also induced a reduction in inflammatory infiltration [27],increased re-epithelization, and improved collagen organization. Similarly, *Abelmoschus esculentus* (L.) (okra) conjugated to AuNPs demonstrated organized collagen fibers and blood vessels at 12 days [27].

Hematoxylin & eosin staining of histological changes shows that after 7 days, lesions treated with AuNP-BP have greater proliferation and organization of collagen fibers and fibroblasts (even more so in AuNP-BP than in BP), with better neovascularization and a reduced presence of inflammatory cells, indicating a lesion in the late reparative phase (granulation tissue formation). On the other hand, in rats without treatment (WOT) or treated with gold nanoparticles (AuNP) alone, there was a greater presence of inflammatory cells in contrast to a lower presence of fibroblasts and collagen. This suggests an earlier stage of injury repair [29].

Masson’s trichomic stain clearly shows the organization of type I collagen (blue), while picrosirius red shows the presence and organization of type III collagen, which is essential at the beginning of the repair process to bring the edges of the wound closer together, promoting its re-epithelization and favoring the regeneration of the basement membrane. However, it is later replaced by type I collagen to reinforce the closure and structure of the injured area [30,31]. 

The histological changes are in accordance with other models where gold nanoparticles with polyphenols such as Curcumin (*Curcuma longa* L.) reduce inflammatory infiltrate, while increasing the collagen area and contraction rate of the wound in a palatine wound model at 5 days [32].

Collagen biosynthesis involves multiple complex steps requiring the temporal and spatial coordination of several biochemical events. Following transcription, the nascent/pre-pro-collagen is post-translationally modified in the endoplasmic reticulum into pro-collagen with the removal of the signal peptide on the N-terminus. Hydroxylation and glycosylation of amino acid residues result in the formation of the triple-helical structure. Supported by chaperone proteins, the pro-collagen triple-helical structure is stabilized for further processing and maturation in the Golgi apparatus and assembled into secretory vesicles that are extruded into the extracellular matrix where the pro-collagen is enzymatically modified into tropocollagen. The final collagen fibril assembly occurs by covalent cross-linking. The mechanical properties of fibrillar collagens are dependent on this cross-linking process, which includes disulfide bonds, both reducible and mature cross-links produced via the lysyl oxidase pathway, transglutaminase cross-links, and advanced glycation end (AGE) products-mediated cross-links [31].

By FTIR spectroscopy, we found a shift on collagen bands amide I and amide II (1651 cm^−1^ and 1549 cm^−1^). These shifts suggest conformational changes in the helices, as analyzed by the increase in the ratio at 1657–1652 cm^−1^/1689–1679 cm^−1^, which showed an increase in AuNP-BP, demonstrating a better collagen alignment, as it was described by de Campos Vidal and Mello, 2019 [20,33]. 

The findings obtained by FTIR spectroscopy are in accordance with Masson’s trichrome and Picrosirius Red staining results, which showed better alignment and improvement in collagen deposition; similar findings were also observed by Chen et al., 2023, in which nepebracteatalic acid nanoparticles improved collagen deposition [34].

The other change observed at 1082 cm^−1^ is related to the symmetric stretching of phosphate in nucleic acids. This change showed an increase in spectral data for both BP and AuNP-BP, suggesting an enhancement in cell proliferation as a result of the improved wound healing process [30].

The increase in this peak is also associated with the increase in fibroblast numbers observed in the H&E staining. As was described, fibroblasts in wound healing (both resident and myeloid cell-converted fibroblasts) are the main source of newly synthesized collagen [31].

The increase in fibroblasts, reduction in the inflammatory process, and the improvement in collagen alignment could be related to the synergistic effect of flavonoids from different traditional medicinal plants, and the effects of AuNP, as was demonstrated by individual analyses [4,5,35].

All results together suggest that nanoconjugation of biological compounds could accelerate the wound healing process, thereby reducing the time required for tissue repair. This leads to the formation of tissue with better physiological and mechanical properties and minimizes the potential adverse side effects produced by some of the bioactive compounds.

New studies currently underway aim to determine whether AuNP-BP treatment can also be used to remodel pre-existing scars resulting from previous injuries. We have also initiated studies to incorporate the formulation into a topical patch that would enable controlled release of the formulation. This approach could potentially be used in patients with chronic wounds of vascular origin, such as diabetic foot, or pressure ulcers, which require prolonged treatment.

## 4. Materials and Methods

### 4.1. Extraction of Polyphenolic Compounds from Bacopa Procumbens

We followed the procedure previously described to obtain the aqueous fraction from *B. procumbens* (Mill.) Green, which was harvested in the state of Hidalgo, Mexico. The whole plant was dried and ground. Then, 40 g of powder was extracted with a 50:50 aquoethanolic solution (600 mL) at 76 °C for 4 h three times using a reflux system. Finally, it was lyophilized. Subsequently, 10 g of this material was resuspended in 100 mL of water for 2 h at room temperature. Afterward, the solvent was removed under vacuum conditions to obtain the aqueous fraction, which was then lyophilized [8]. 

### 4.2. Gold Nanoparticles Synthesis

Gold nanoparticles were prepared by the chemical reduction of tetrachloroauric acid trihydrate (Sigma-Aldrich, St. Louis, MO, USA) with sodium citrate dehydrate (Sigma-Aldrich, St. Louis, MO, USA) in water. The AuNPs were synthesized because the citrate ions act as both reducing and capping agents. This method involved preparing 1 mL of HAuCl4 (Sigma-Aldrich, St. Louis, MO, USA) at 4% in deionized water. A quantity of 0.5 mL of this solution was added to 200 mL of deionized water and heated to boiling with constant stirring. Once the temperature reached between 97 and 100 °C, 3 mL of 1% sodium citrate was added. Subsequently, the solution began to darken, turning bluish-gray or purple. After 30 min, the reaction was complete, and the final deep wine-red color indicated the formation of a colloidal dispersion of gold nanoparticles. After cooling, the dispersion was centrifuged at 3500 rpm for 40 min. The supernatant was then removed, and the nanoparticles were resuspended in 6 mL of deionized water. The resulting suspension was stored at 4 °C [13].

### 4.3. Gold Nanoparticles Conjugation with Polyphenolic Compounds of Bacopa Procumbens

Once synthesized, the AuNPs with negative charges provided by the citrate groups located on their surface, were mixed with different concentrations of polyphenolic compounds of *B. procumbens* (BP) to cover the surface of nanoparticles with a BP extract layer. For this purpose, constant volumes of the colloidal dispersion nanoparticles (AuNPs) were mixed with constant volumes of the BP at several concentrations (0.4, 0.8, 1.6, 3.2, and 6.4 mg/mL) overnight, after the conjugates were cleaned by centrifugation twice at 4 °C, 1000 rpm during 10 min. The resulting conjugate nanoparticles (AuNP-BP) were then characterized.

### 4.4. Instrumentation

The solution in which the nanoparticles were suspended was dried. Briefly, both AuNPs and AuNPs-BP were carefully deposited onto carbon-coated TEM grids, which have a 300 mesh with copper and approximate hole sizes of 63 µm (Waltham, MA, USA). High-resolution microphotographs were obtained using the HRTEM mode on a transmission electron microscope (TEM, model JEM2100 with a LaB6 emitter) (JEOL, Peabody, MA, USA). ImageJ software (1.54) was used to determine the size of AuNPs and AuNPs-BPs

Ultraviolet-visible measurements were performed using an Evolution 606 Spectrophotometer (Thermo Fisher Scientific, Waltham, MA, USA). This instrument was utilized to measure the surface plasmon resonance (SPR) absorption band of single gold nanoparticles (AuNPs), polyphenolic compounds of *B. procumbens* (BP), and the conjugate (AuNP-BP) [23]. 

A Fourier Transform Infrared (FTIR) spectrometer, Bruker model Vertex 70 in the attenuated total reflectance (ATR) sampling mode, was used to measure the infrared absorptions of AuNPs, BP, and AuNP-BP at 1.6 mg/mL. For the FTIR measurements, the colloidal samples (AuNPs and AuNP-BP) were centrifuged at 3500 rpm for 40 min; then, the supernatant was removed, and 2 μL of the concentrated sample was placed on the surface of the ATR crystal. The infrared radiation was propagated along the crystal to obtain the corresponding vibrational spectrum, which was averaged from multiple data acquisitions. The infrared spectra were collected in the biological fingerprinting range of 1800 to 800 cm^−1^. 

Zetasizer Nano ZS (Malvern Instruments Ltd., Malvern, UK) Dynamic Light Scattering (DLS) and Electrophoretic Light Scattering (ELS) techniques were used to analyze the size and the zeta potential of the AuNPs and the AuNP-BP samples after dilution with ultrapure distilled water [26]. 

### 4.5. In Vivo Skin Wound Model

#### 4.5.1. Animals

Twenty four male adult Wistar rats (220–280 g) were maintained at 26 °C under 12:12 h light/dark cycle. Animal received chow and water ad libitum and were kept in individual cages. The ethics committee of ENMyH postgraduate section approved the experimental procedure of this study (CBE/010/2019) [8]. 

The animals were divided into the following four randomized groups: untreated group (WOT), AuNPs at 5% (AuNPs), group treated with polyphenolic compounds of *B. procumbens* (BP) at 160 mg/mL, and AuNP-BP at 1.6 mg/mL, all of them included in hydrogel. For hydrogel integration, in a reactor equipped with stirring blades, 70% deionized water was set under stirring at minimal velocity. To this, AuNP-BP, 1% glycerin, hydantoin, 0.2% methylchloroisothiazolinone, and methylisothiazolinone were added gradually over 10 min. Then, 0.7% carbopol was introduced, with continuous stirring for an additional 10 min. Finally, 1% triethanolamine was added until gel formation was achieved. The mixing process was then maintained at medium velocity for 15 min.

For the excisional wound model, the animals were anesthetized intraperitoneally with ketamine/xylazine (50/5 mg/kg). One day prior to creating the wounds, the dorsal region of each animal was depilated. Subsequently, four full-thickness excisional wounds, each measuring 1 cm^2^, were created. The wounds were treated daily with 100 μL from the different treatments, wounds were not covered with any other film. Animals were sacrificed on day 7 after injury. Six rats were used for each experimental group [8].

#### 4.5.2. Morphometric Analysis

The healings were observed, photographed, and measured immediately after surgery (initial wound area) and on day 7 using a Vernier caliper to calculate the percent reduction of wounds according to Equation (1) [8].
% of wound reduction = 100 − [(final area × 100)/initial area](1)

#### 4.5.3. Histopathological Analysis

Wound lesions from the different groups on day 7 were surgically removed along with adjacent healthy skin. Tissues were then fixed in 4% buffered paraformaldehyde, embedded in paraffin, and sectioned using a microtome. Five µm thick sections were stained with Hematoxylin & Eosin, Masson’s trichrome stain, or PicroSirius Red reagent (Abcam, Cambridge, UK), according to the manufacturer’s instructions. Subsequently, the tissues were observed and photographed using the Olympus DP74 system (Olympus, Tokyo, Japan) and a polarizing light microscope (Nikon, Tokyo, Japan). Three different areas from three consecutive sections were analyzed for descriptive histopathological analysis [8].

#### 4.5.4. ATR-FTIR Spectra Analysis

The biomolecular and structural analysis of wounds was performed through vibrational spectroscopy. The samples were placed directly on the surface of the ATR crystal for the acquisition of their FTIR spectra; WOT, AuNPs, BP, and AuNP-BP were analyzed by FTIR as described previously in the instrumentation section.

Spectral analysis was conducted in the fingerprint region (1800–800 cm^−1^) using OPUS software (version 7.1, Bruker Daltonics GmbH & Co. KG, Bremen, Germany). Absorbance spectra were normalized using a standard normal variate (SNV) normalization, employing the Unscrambler X software (version 10.3, Camo). The bands related to lipids, proteins, collagen, and phosphates were identified in each spectrum, and the average spectra were calculated. The frequencies of the bands and the absorbances of the spectra were analyzed using Origin software (version 6.1, Origin Lab Corporation, Northampton, MA, USA). Then, the second derivative was calculated employing the Savitzky-Golay algorithm with a five-point window for smoothing and the second polynomial order using Unscrambler X. The second derivative is a mathematical tool that allows for the observation of absorption bands related to the secondary structure of proteins. In this case, the secondary structures (α-helix and β-sheet) related to several types of collagens in the Amide I region (1700–1600 cm^−1^) were observed [22]. 

The integrate area corresponding to α-helix (1657–1652 cm^−1^) and β-sheet (1689–1679 cm^−1^) in the second derivative was obtained to calculate the ratio of α-helix/β-sheet [20].

### 4.6. Statistical Analysis

Statistical significance was analyzed using either the ANOVA post hoc Turkey’s test or Kruskal-Wallis test, according to the distribution of the data. All analyses were performed using Graph Pad Prism software, version 8.0.

## 5. Conclusions

The formulation and nanostructuring of polyphenolic compounds from *B. procumbens* with AuNPs of an average size of 16.5 nm effectively promote healing by reducing by 100 times the concentration of BP required for tissue regeneration. The nanoformulation (AuNP-BP) applied topically accelerated wound healing, resulting in an 85.2% reduction in the lesion size after 7 days. Histological changes showed a significant decrease in the inflammatory phase, increasing fibroblast proliferation, and an improvement in collagen organization. It induced structural changes in collagen, as evidenced by FTIR spectra, increasing the α-helix/β-sheet ratio from 1 in animals without treatment to 1.22 after topical treatment with the nanoformulation (AuNP-BP).

## 6. Patents

The patent derived from this paper and previous work is registered with number 376170 [8].

## Figures and Tables

**Figure 1 ijms-24-17138-f001:**
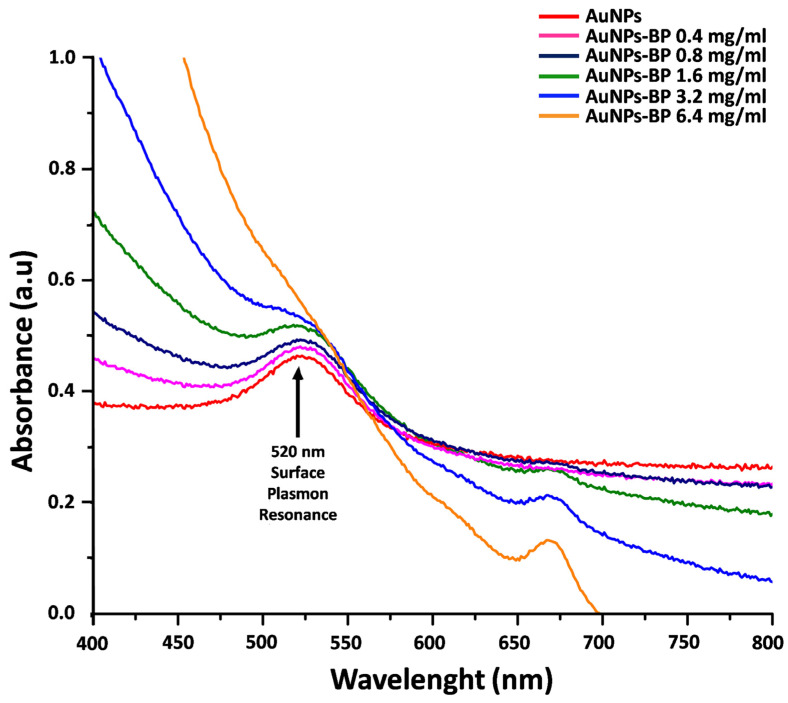
UV-visible spectra of AuNPs and AuNP-BP conjugate at different concentrations of the polyphenolic compounds (BP) from *B. procumbens* (0.4–6.4 mg/mL).

**Figure 2 ijms-24-17138-f002:**
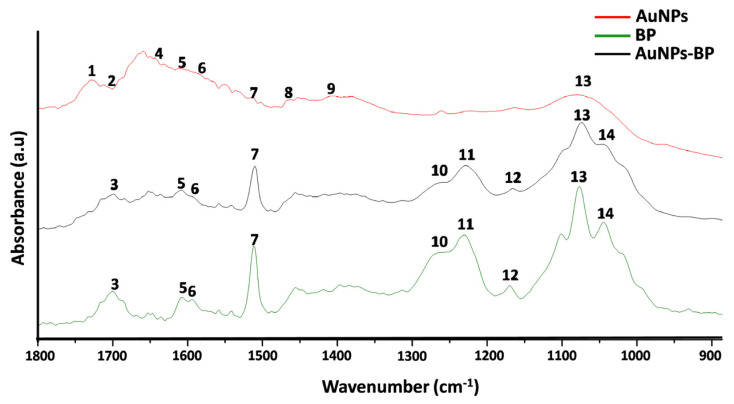
FTIR spectra of AuNPs, BP, and AuNP-BP at 1.6 mg/mL. Band frequencies marked as 1 to 14 are described in Table 1.

**Figure 3 ijms-24-17138-f003:**
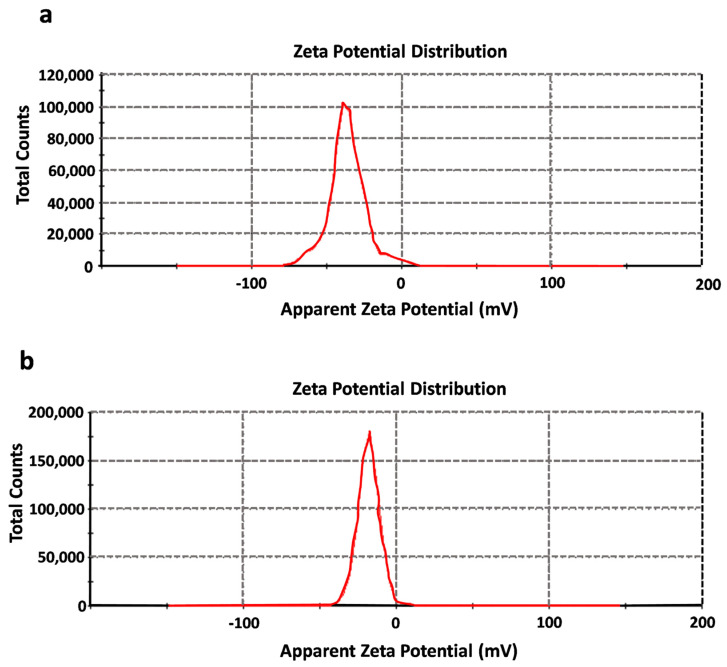
Z potential of (**a**) AuNPs and (**b**) AuNP-BP at 1.6 mg/mL.

**Figure 4 ijms-24-17138-f004:**
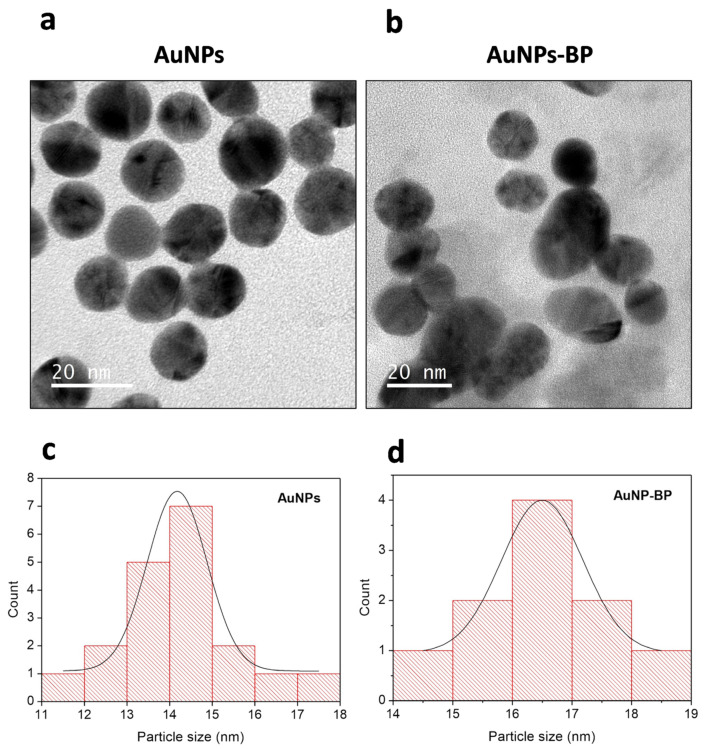
Transmission Electron Microscopy images of AuNP and AuNP-BP (**a**,**c**), accompanied by size distribution graphs showing an average particle diameter of 14.18 nm for AuNP, and 16.5 nm for AuNP-BP (**b**,**d**).

**Figure 5 ijms-24-17138-f005:**
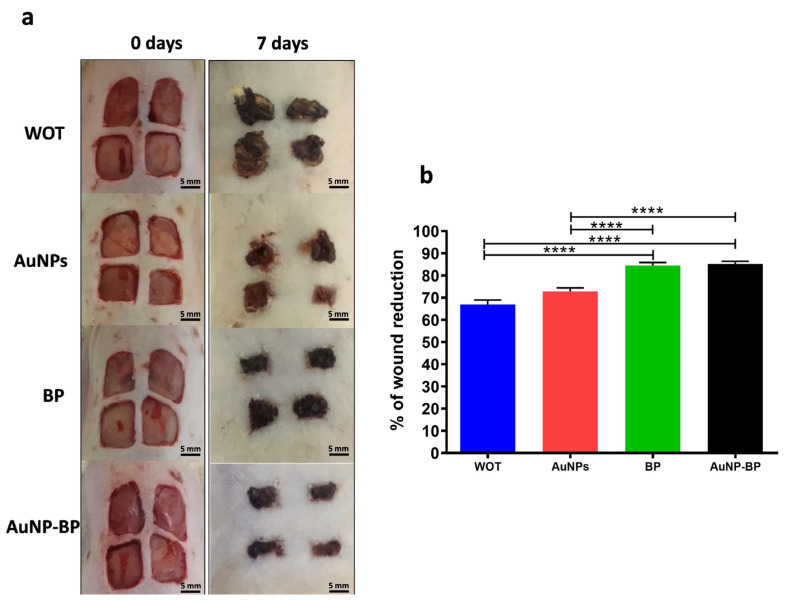
Morphometric changes induced by the different topical treatments. (**a**) Macroscopical changes in groups at 0 and 7 days post wound healing. (**b**) Wound area reduction between groups. **** *p* < 0.0001. WOT, without treatment group; AuNPs, gold nanoparticles group; BP, *B. procumbes* polyphenolic compounds group; AuNP-BP, gold nanoparticles conjugated with BP group. n = 24 wounds per group.

**Figure 6 ijms-24-17138-f006:**
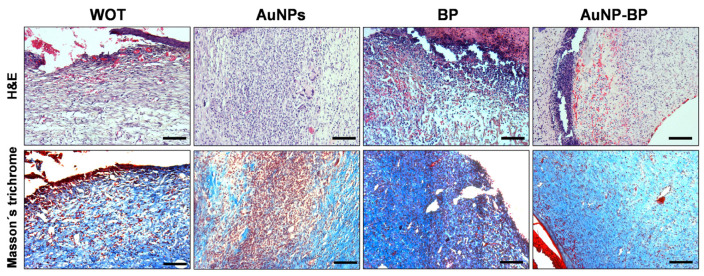
Histological changes of wounds after 7 days of treatment. Representative microphotographs of wounds stained with Hematoxylin/Eosin (H&E) and Masson’s trichrome in WOT, AuNPs, BP, and AuNP-BP. Scale Bars = 100 µm.

**Figure 7 ijms-24-17138-f007:**
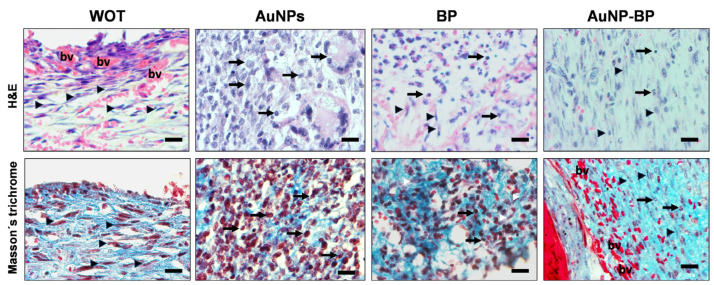
Amplifications of microphotographs shown in Figure 6 stained with Hematoxylin/Eosin (H&E) and Masson’s trichrome. The images highlight the cell populations associated with tissue repair in WOT, AuNPs, BP, and AuNP-BP groups. Inflammatory cells (arrows); fibroblasts (arrowheads); blood vessels (bv). Scale Bars = 20 µm.

**Figure 8 ijms-24-17138-f008:**
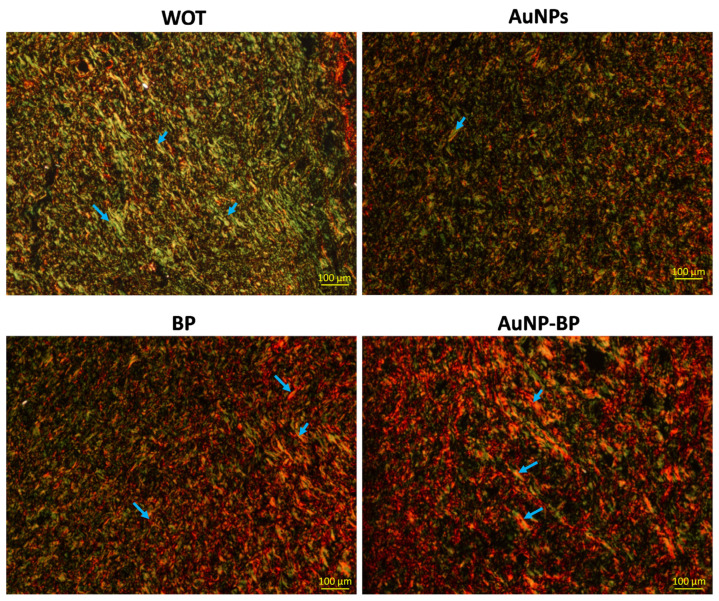
Collagen organization on wounds after 7 days of treatment. Representative microphotographs of Picrosirius Red staining under polarized light in WOT, AuNPs, BP, and AuNP-BP groups. Greenish fibers (collagen type III) and yellow-red fibers (collagen type I). Collagen fibers (blue arrow). Scale Bar = 100 µm.

**Figure 9 ijms-24-17138-f009:**
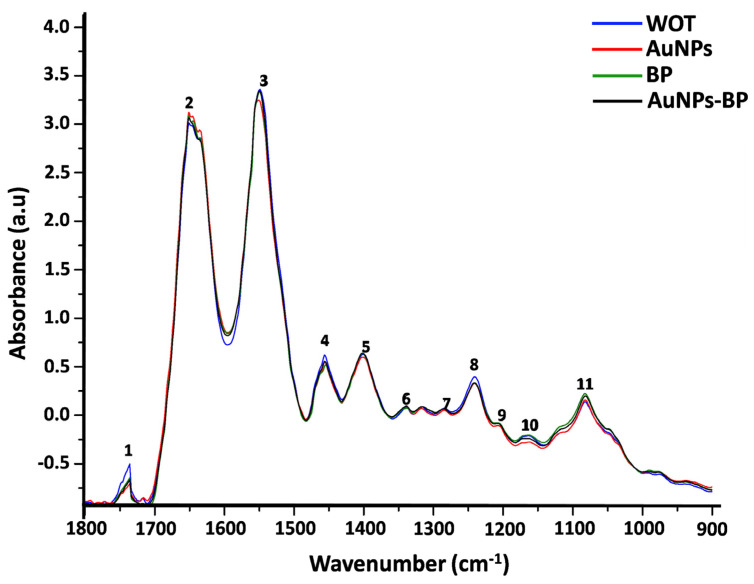
FTIR spectra average of WOT, AuNPs, BP, and AuNP-BP groups. Band frequencies marked as 1 to 11 are described in Table 2.

**Figure 10 ijms-24-17138-f010:**
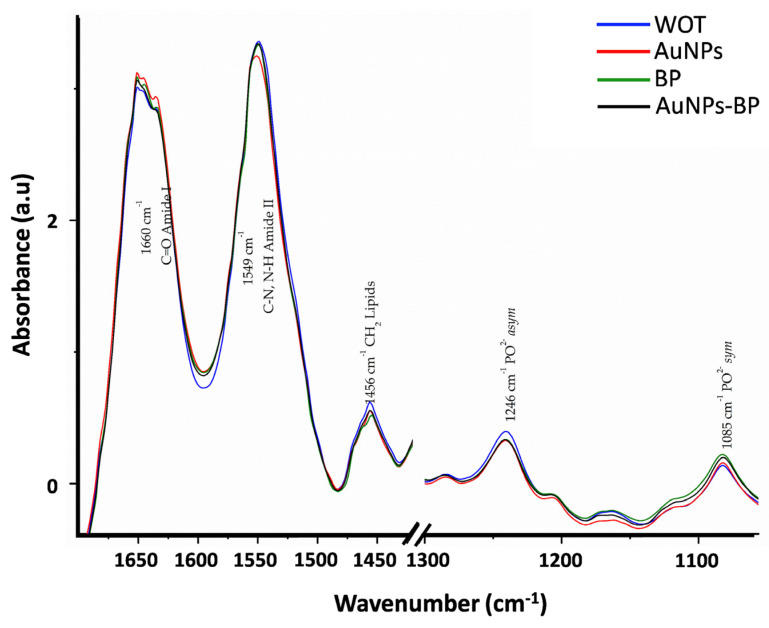
Bands with significant major peak shift. FTIR spectra of skin after 7 days of healing; without treatment (WOT), and with different treatments: AuNPs, BP, and AuNP-BP.

**Figure 11 ijms-24-17138-f011:**
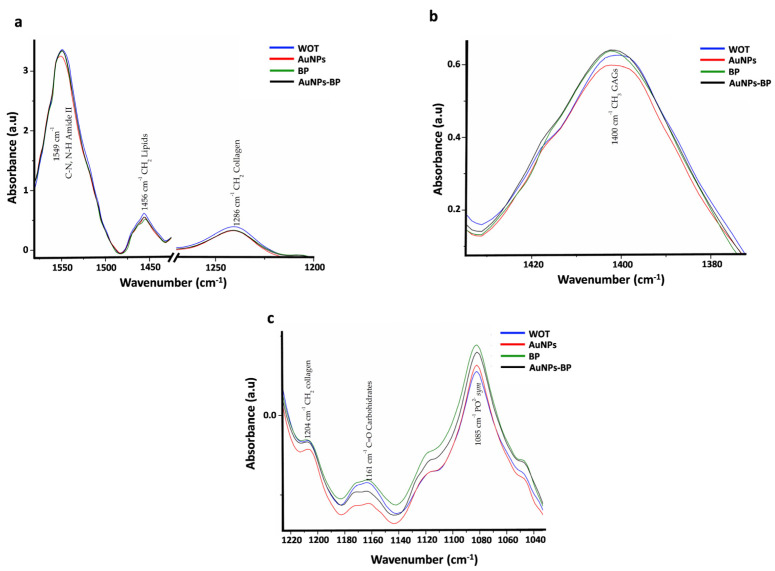
Bands with significant changes in absorbance. (**a**) Bands with a significant decrease in the BP and AuNP-BP groups compared to WOT or AuNPs. (**b**) Bands that increased in BP and AuNP-BP groups in comparison with AuNPs. (**c**) Bands with an increase compared to AuNPs.

**Figure 12 ijms-24-17138-f012:**
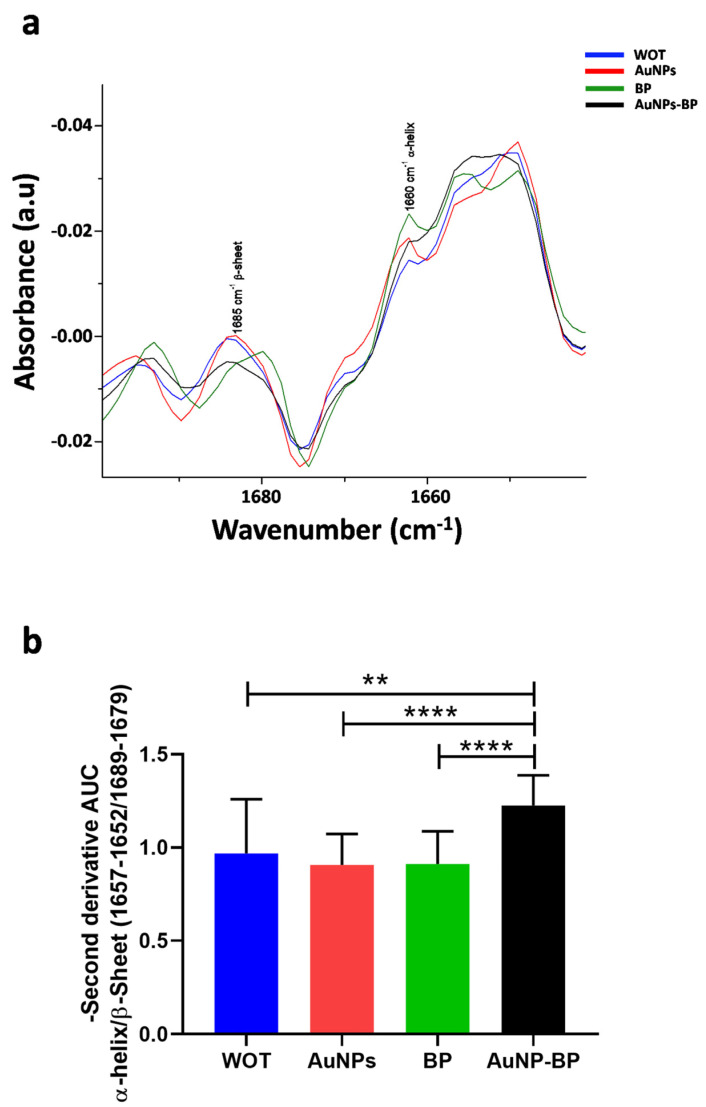
Negative of Second derivative FTIR spectra of Amide I. (**a**) FTIR spectra average of negative second derivative of WOT, AuNPs, BP, and AuNP-BP groups. (**b**) AUC ratio α-helix/β sheet. ** *p* < 0.01, **** *p* < 0.0001. n = 18 measurements per group.

**Table 2 ijms-24-17138-t002:** FTIR bands assignment.

Band	cm^−1^	Assignment	References
1	1737	C=O lipids	[22]
2	1660	C=O Amide I	[21,22]
3	1549	C-N, N-H Amide II	[21,22]
4	1456	CH_2_ lipids	[21,22]
5	1400	CH_3_ GAGs	[21]
6	1338	CH_2_ Collagen type I	[21]
7	1286	CH_2_ collagen Amide III, glycine and proline	[21]
8	1246	PO_2_^−^ *_asym_* Phospholipds	[22]
9	1204	CH_2_ Collagen Amide III	[21]
10	1161	C-O carbohydrate residues	[21]
11	1085	PO_2_^−^ *_sym_* Phospholipds, C-O of carbohydrate on Collagen and PG	[21,22]

**Table 3 ijms-24-17138-t003:** FTIR band frequencies observed in skin at 7 days on groups without treatment (WOT), or treated with AuNPs, BP, and AuNP-BP.

Band	WOT cm^−1^ Media (IQR)	AuNPs cm^−1^ Media (IQR)	BP cm^−1^ Media (IQR)	AuNP-BP cm^−1^ Media (IQR)	*p*
1	1736 (1736, 1736)	1736 (1736, 1736)	1736 (1736, 1736)	1736 (1736, 1736)	0.069
2	1651 (1645, 1651)	1651 (1651, 1651) ^a^	1651 (1651, 1651) ^a^	1651 (1651, 1651) ^a^	0.016
3	1549 (1549, 1549)	1551 (1551, 1552) ^a^	1551 (1549, 1551) ^a^	1551 (1549, 1551) ^ab^	0.000
4	1456 (1454, 1456)	1456 (1456, 1456)	1456 (1454, 1656) ^b^	1456 (1456, 1456.)	0.001
5	1402 (1400, 1402)	1402 (1402, 1402)	1402 (1402, 1402)	1402 (1402, 1402)	0.317
6	1339 (1339, 1340)	1339 (1339, 1340)	1339 (1337, 1340)	1340.46 (1339, 1340)	0.713
7	1285 (1283, 1286)	1285 (1285, 1286)	1286 (1285, 1288)	1285 (1285, 1286)	0.081
8	1240 (1240, 1240)	1240 (1240, 1242)	1240 (1240, 1242)	1242 (1240, 1242) ^ab^	0.028
9	1207 (1207, 1211)	1207 (1207, 1209)	1209 (1207, 1211)	1209 (1207, 1211)	0.088
10	1163 (1161, 1171)	1164 (1162, 1168)	1163 (1163, 1171)	1167 (1163, 1171)	0.497
11	1082 (1082, 1082)	1082 (1082, 1082)	1082 (1082, 1084) ^ab^	1082 (1082, 1082)	0.000

p Kruskal-Wallis test. ^a^ *p* <0.05 vs. WOT. ^b^ *p* <0.05 vs. AuNPs. IQR: Interquartile Range.

**Table 4 ijms-24-17138-t004:** FTIR bands absorbance observed in skin at 7 days for groups without treatment (WOT), treated with gold nanoparticles (AuNPs), BP, and conjugate (AuNP-BP).

Band	WOT Absorbance Media (IQR)	AuNPs Absorbance Media (IQR)	BP Absorbance Media (IQR)	AuNP-BP Absorbance Media (IQR)	*p*
1	0.4 (0.16, 0.64)	0.19(0.12, 0.29)	0.28 (0.23, 0.62)	0.28 (0.23, 0.34)	0.08
2	4.03 (3.88, 4.17)	4.08 (4.96, 4.13)	4.09 (4.06, 4.2)	4.07 (4.03, 4.1)	0.128
3	4.36 (4.33, 4.45)	4.21 (4.19, 4.24) ^a^	4.35 (4.24, 4.45) ^b^	4.28 (4.24, 4.33) ^ab^	0.0001
4	1.58 (1.52, 1.74)	1.51 (1.47, 1.54)	1.55 (1.51, 1.61) ^b^	1.53 (1.49, 1.55) ^a^	0.012
5	1.64 (1.55, 1.66)	1.56 (1.53, 1.58) ^a^	1.63 (1.58, 1.74) ^b^	1.63 (1.56, 1.67) ^b^	0.001
6	1.06 (1.01, 1.13)	1.06 (1.03, 1.08)	1.09 (1.04, 1.15)	1.06 (1.05, 1.08)	0.305
7	1.06 (1, 1.12)	1.01 (0.98, 1.04) ^a^	1.07 (0.99, 1.17) ^b^	1.02 (1.02, 1.04)	0.04
8	1.36 (1.28, 1.44)	1.3 (1.23, 1.34)	1.36 (1.25, 1.48)	1.28 (1.26, 1.33)	0.091
9	0.89 (0.84, 0.99)	0.85 (0.81, 0.89) ^a^	0.93 (0.84, 1.02) ^b^	0.89 (0.86, 0.93) ^b^	0.015
10	0.76 (0.72, 0.87)	0.66 (0.63, 0.72) ^a^	0.79 (0.69, 0.92) ^b^	0.75 (0.69, 0.81) ^b^	0.003
11	1.14 (0.99, 1.22)	1.11 (1.07, 1.17)	1.19 (1.13, 1.35) ^ab^	1.2 (1.16, 1.23) ^b^	0.003

p Kruskal-Wallis test. ^a^ *p* < 0.05 vs. WOT. ^b^ *p* < 0.05 vs. AuNPs. IQR: Interquartile Range.

## Data Availability

The data that support the findings of this study are available on request from the corresponding author.

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
