# Peer review of "Polyphenolic Compounds Nanostructurated with Gold Nanoparticles Enhance Wound Repair"

_ijms, 2023, doi:10.3390/ijms242417138_

Round 1

Reviewer 1 Report

Comments and Suggestions for Authors

Reviewer comments to the Authors:

 The first paragraph of the introduction should be supported with appropriate references to provide a solid foundation for the study's context.

 Clarify the family of Bacopa procumbens in the introduction.

 Throughout the manuscript, ensure consistency by using "UV" instead of "Uv."

 In the methodology, specify which part of Bacopa procumbens was used for the study to provide more detailed information.

 The scientific name of Bacopa procumbens should be italicized throughout the manuscript for correct scientific nomenclature.

 Double-check the spelling of "Xylazine" for accuracy.

 The sentence stating, "four complete thickness excision surgeries of 1 cm2 were performed," is unclear. Please specify whether the author means four wounds were created, each measuring 1 cm2. If not, provide the correct size of each wound.

 Explain the meaning of "dorsal region was harvested" to ensure clarity.

 Include the size of the thickness sections in the histology section for completeness.

 Mention the use of Masson's trichrome staining in the methodology, as it appears to be missing.

 In the conclusion section, consistently refer to "bacopa" as "Bacopa" for clarity and scientific accuracy.

 Clarify whether Tegaderm or any other film was applied to cover the wounds.

 Explain the rationale for terminating the study on Day 7 and not continuing until complete wound healing. Discuss the potential benefits of extending the study duration to compare the effect of AuNP-BP with control wounds.

 In Figure 4, scab formation can be seen, which hinders the wound area calculation in the early stage. (For example, compare the size of the 4th Wound vs the 2nd wound treated with AuNPs. Also, it is good to include the wound images of Day 0, and Day 5 to compare the wound size.

 Add a scale bar to each histology image in Figure 5 to provide a reference for size.

 Correct "Masson" to "Masson's trichrome" in Figure 5.

 If normal skin staining is included in H&E and MT Staining, explicitly mention it. If not, exclude the normal skin staining from Figure 6 to have uniformity.

 Expand the discussion section by comparing the study's efficacy with other relevant studies, not limited to Abelmoschus esculentus. Additionally, discuss the histology data, especially the H&E staining results.

 While the chemistry part of the study is well-described, prioritize discussing the wound healing aspects and histology data using existing literature for a more comprehensive analysis.

 Include a section discussing the limitations of the study to provide a well-rounded evaluation.

 Mention the future directions of the study to indicate possible avenues for further research and development.

Author Response

Reviewer 1

Comment. -The first paragraph of the introduction should be supported with appropriate references to provide a solid foundation for the study's context.

Answer. – Thanks, we added references related to the wound healing process. (lines 557-562)

Comment. -Clarify the family of Bacopa procumbens in the introduction.

Answer. – Thank you for your observation. We have incorporated suggested information on Bacopa procumbens, a member of Scrophulariaceae family (please see lines 61-62).

Comment. -Throughout the manuscript, ensure consistency by using "UV" instead of "Uv."

Answer. – We have homogenized the term “UV” throughout the manuscript.

Comment. -In the methodology, specify which part of Bacopa procumbens was used for the study to provide more detailed information.

Answer. –  Thank you for your comment. We have added a new paragraph on the methodology, which can be found in lines 408-413:

“plant was harvested on the state of Hidalgo, Mexico, then whole plant was dried and ground, 40 g of powder was extracted with aquoethanolic solution 50:50 (600 mL) at 76 C for 4 h three times using a reflux system, and finally it was lyophilized, then 10 g was resuspending on 100 ml of water, for 2 h at room temperature, after solvent was removed under vacuum conditions to obtain aqueous fraction then it was lyophilized”.

Comment. - The scientific name of Bacopa procumbens should be italicized throughout the manuscript for correct scientific nomenclature.

Answer. –  Thank you for the observation. All scientific names were revised and wrote in italics.

Comment. -Double-check the spelling of "Xylazine" for accuracy.

Answer. –  The name was corrected on line 479.

Comment. -The sentence stating, "four complete thickness excision surgeries of 1 cm2 were performed," is unclear. Please specify whether the author means four wounds were created, each measuring 1 cm2. If not, provide the correct size of each wound.

Answer. – Thanks for the observation, we rewrote that paragraph, please see lines 478-481: “For the excisional wound model, the animals were anesthetized intraperitoneal with ketamine/xylazine (50/5 mg/kg), one day previous to the wounds, the dorsal region was depilated, and four full-thickness excisional wounds were created, each measuring 1 cm2. The wounds were treated daily with 100 μl from the different treatments. Animals were sacrificed at day 7 after injury. Six rats were used for each experimental group”.

Comment. -Explain the meaning of "dorsal region was harvested" to ensure clarity.

Answer. – We agree with the observation and thank you for it. To enhance clarity, we have improved the wording in the relevant section. Please see the content on line 480.

Comment. -Include the size of the thickness sections in the histology section for completeness.

Answer. – We added the information as: Five µm thickness sections were stained please see line 492, thanks.

Comment. -Mention the use of Masson's trichrome staining in the methodology, as it appears to be missing.

Answer. – Thank you, we are so sorry, we already added it in the methodology, line 493.

Comment. -In the conclusion section, consistently refer to "bacopa" as "Bacopa" for clarity and scientific accuracy.

Answer. – Thank you, we already did, thanks.

Comment. -Clarify whether Tegaderm or any other film was applied to cover the wounds.

Answer. – For these experiments we did not use Tegaderm film or any other film (line 482)

Comment. -Explain the rationale for terminating the study on Day 7 and not continuing until complete wound healing. Discuss the potential benefits of extending the study duration to compare the effect of AuNP-BP with control wounds.

Answer. – Thank you for your comment, we are interested in evaluating the changes induced by AuNP-BP compared to BP in the early and intermediate phases of the repair process, where the processes related to proliferation, reduction of inflammation and reorganization of collagen occur, as we have shown previously (Martinez-Cuazitl et al 2022). We have also added FTIR analysis, which has allowed us to learn new aspects of these initial phases. In the final phase of the process the most important aspects of the healing process have already completed. We have previously shown (Martinez-Cuatzitl et al 2022), that after 8 or 9 days the process fundamentally involved in the remodelling of the extracellular matrix. However, as you suggested it would be of interest for the next work, to study the effect of the nanoconjugate to demonstrated if it can remodel the extracellular matrix in scars already generated. In discussion we already included a paragraph about the potential benefits to characterize the effect in late phases. (please see lines 399-404).

Comment. -In Figure 4, scab formation can be seen, which hinders the wound area calculation in the early stage. (For example, compare the size of the 4th Wound vs the 2nd wound treated with AuNPs. Also, it is good to include the wound images of Day 0, and Day 5 to compare the wound size.

Answer. – Thanks, is very pertinent your comment. We added the representative images at day 0 (please see figure 5), the scab was not considered for the measurement, we performed the measures at the base of the wound area at day 7 after euthanized to take the real size. On that sense we could not have the measure at day 5.

Comment. -Add a scale bar to each histology image in Figure 5 to provide a reference for size.

Answer. – Thanks, the scale bars were added now in Figure 6 and we added new image to improve the quality of the histological analysis, now Figure 7.

Comment. -Correct "Masson" to "Masson's trichrome" in Figure 5.

Answer. – Thank you, we correct the misspelling (lines 206 and 210 and Figures 6 and 7).

Comment. -If normal skin staining is included in H&E and MT Staining, explicitly mention it. If not, exclude the normal skin staining from Figure 6 to have uniformity.

Answer. – Thak you, we exclude the normal skin to have uniformity and now due we added 2 figures, the figure is figure 8. (line 218)

Comment. -Expand the discussion section by comparing the study's efficacy with other relevant studies, not limited to Abelmoschus esculentus. Additionally, discuss the histology data, especially the H&E staining results.

Answer. – Thank you, we added other studies to compare their results in comparison with our results related to the histology findings. Please see lines 336-358.

Comment. -While the chemistry part of the study is well-described, prioritize discussing the wound healing aspects and histology data using existing literature for a more comprehensive analysis.

Answer. – Thank you, we already extended the wound healing and histology data in the discussion section. Please see lines 336-372.

Comment. -Include a section discussing the limitations of the study to provide a well-rounded evaluation. Mention the future directions of the study to indicate possible avenues for further research and development.

Answer. – Thanks, we already add a paragraph mentioning the new studies and future directions of our research. Please see lines 399-404. New studies currently underway will allow us to determine whether AuNP-BP treatment can also be used to remodel pre-existing scars resulting from previous injuries. We have also initiated studies to incorporate the formulation into a topical patch that can control the release of the formulation and can be used in patients with chronic wounds of vascular origin, diabetic foot or pressure ulcers that require prolonged treatment.

Reviewer 2 Report

Comments and Suggestions for Authors

This paper presented a conjugate of BP molecule and Au NP for wound healing applications. The idea is novel and the benefit of reducing dosing of BP by conjugating with AuNP is significant in the nanomedicine field. However, the characterization of the conjugate NP should be improved before considering acceptance. 

1. TEM image should be provided to confirm the particle size. 

2. The mechanism of binding of BP and Au is not clear. From IR, there is no difference between bound BP and free BP. it just physisorption on the surface? How stable is the binding in practical medical situations? 

3. The exact amount of BP bound on the surface should be provided. 

Author Response

This paper presented a conjugate of BP molecule and Au NP for wound healing applications. The idea is novel and the benefit of reducing dosing of BP by conjugating with AuNP is significant in the nanomedicine field. However, the characterization of the conjugate NP should be improved before considering acceptance.

Comment. TEM image should be provided to confirm the particle size. 

Answer. – Thanks for your suggestion, images of the gold nanoparticles (AuNPs) used to prepare the BP conjugate were obtained by transmission electron microscopy (TEM). They have been included in the manuscript for a better appreciation of their icosahedral morphology. The average particle size obtained by image processing using Image J software was 14.18 nm for AuNP and 16.5 nm for AuNP-BP. (Please see lines 155-168, 438-443).

Comment. The mechanism of binding of BP and Au is not clear. From IR, there is no difference between bound BP and free BP. it just physisorption on the surface? How stable is the binding in practical medical situations? 

Answer. –  Thank you for your comments and questions. Since AuNPs were synthesized by chemical reduction with sodium citrate, at the end of the synthesis, they are covered on their surface by the citrate ion, which confers a negative charge to the particles. This electronegativity of the gold nanoparticles generates an electrostatic attraction to the multiple antioxidant compounds present in the B. procumbens extract during the conjugation process. This explains why the FTIR spectrum of the BP-conjugate is observed to be very similar to the spectrum of the BP extract. On the one hand, gold nanoparticles exhibit the surface plasmon resonance phenomenon, which is responsible for their optical properties, allowing their size, shape, and composition to be tracked and easily controlled. (Please see lines 267-275).

Comment. The exact amount of BP bound on the surface should be provided.

Answer. –  Thanks, we already calculated it and include in the manuscript, please see lines 293-300. Higher concentrations of AuNP-BP displayed a peak at 668 nm, related to the chlo-rophyll α. Some reports have shown that higher concentrations of extracts could act as capping and stabilizing agents, such as phenolics, flavonoids, alkaloids, polysaccharides, saponins, tannins, and organic acids, leading to Au-hydro-complex formation rather than nanoparticles. Thus, we chose a BP concentration of 1.6 mg/mL for optimal capping formation, while the concentration of nanoparticles, estimated by the Haiss method through the UV-vis spectrum, was 3.83x10¹³ particles/ml. Therefore, each particle contains 4.18 ag/p of BP extract on its surface.

Reviewer 3 Report

Comments and Suggestions for Authors

- The title does not need to have abbreviations - the '(AuNP-BP)' should be removed from the title.

- Images in Figure 4 are missing scale bars. Also, the morphological features are not really clear/visible in these low mag images. Authors should provide higher mag insets to clearly display the features for each group.

- The wound healing quantification in Figure 4 is only conducted for 7 days, which is a rather short period for wound regenerative processes to complete. Authors should present longer time periods to monitor closely how these processes continue. Also, important missing info here is the images of the wound at day 0 (baseline), so that the readers can readily compare different treatment conditions.

- Histology results shown in Figure 5: again, missing scale bars. These images need annotation tools to highlight key features. The current form is pretty hard for the readers to follow and fully understand. Also, the control day 0 images are missing here as well.

- Figure 6: it is impossible to understand what the arrows are point at in these images. Very poor quality and indiscernible image quality makes these photos not useful. Enhanced brightness and higher mag photos are needed. The scale bars also not readable.

- For all figures, statistical analysis markers should be defined in the caption. Also, report the sample size (n) for all quantitative data.

Comments on the Quality of English Language

The quality of the English language is OK.

Author Response

Comment - The title does not need to have abbreviations - the '(AuNP-BP)' should be removed from the title.

Answer. – Thank you, it was removed

Comment - Images in Figure 4 are missing scale bars. Also, the morphological features are not really clear/visible in these low mag images. Authors should provide higher mag insets to clearly display the features for each group.

Answer. – Thanks for the observation, we added the representative images at day 0, the scab was not considered for the measurement, we performed the measures at the base of the wound area at day 7 after euthanized to take the real size. On that sense we could not have the measure at day 5 (Please see line 182).

Comment - The wound healing quantification in Figure 4 is only conducted for 7 days, which is a rather short period for wound regenerative processes to complete. Authors should present longer time periods to monitor closely how these processes continue. Also, important missing info here is the images of the wound at day 0 (baseline), so that the readers can readily compare different treatment conditions.

Answer. –  Thank you for your comment, we added the representative images at day 0 (please see figure 5 on line 182). In reference to your other comment, we are interested in evaluating the changes induced by AuNP-BP compared to BP in the early and intermediate phases of the repair process, where the processes related to proliferation, reduction of inflammation and reorganization of collagen occur, as we have shown previously (Martinez-Cuazitl et al 2022). We have also added FTIR analysis, which has allowed us to learn new aspects of these initial phases. In the final phase of the process have completed the most important aspects of the healing process and as we have shown in the previous study, the process fundamentally involved in the remodelling of the extracellular matrix. However, as you suggested it would be of interest for the next work, to study the late phase to evaluate the effect of the nanoconjugate in remodelling the extracellular matrix in scars already generated.

Comment - Histology results shown in Figure 5: again, missing scale bars. These images need annotation tools to highlight key features. The current form is pretty hard for the readers to follow and fully understand. Also, the control day 0 images are missing here as well.

Answer. – Thanks, the scale bars were added at image 5 and we added new image to improve the quality histology analysis. Please see figures 6 and 7 on lines 204-212.

Comment - Figure 6: it is impossible to understand what the arrows are point at in these images. Very poor quality and indiscernible image quality makes these photos not useful. Enhanced brightness and higher mag photos are needed. The scale bars also not readable.
Answer. – Thanks, we already change the image, Figure 8 (line217), were we included the new magnification images precising what the arrows point in the images.

Comment - For all figures, statistical analysis markers should be defined in the caption. Also, report the sample size (n) for all quantitative data.

Answer. – Thak you, we added the markers and the sample size, please see Figure 5 at 187, Figure 12 lines 259-262.

Reviewer 4 Report

Comments and Suggestions for Authors

In this investigation, aimed at evaluating the influence of AuNPs on enhancing the bioactivity of B. procumbens in tissue repair for skin injuries, and given the significance of biocompatibility and the extensive study of AuNPs, the authors undertook the synthesis and characterization of gold nanoparticles functionalized with polyphenolic compounds from B. procumbens (AuNPs-BP). Subsequently, the resulting colloidal dispersion was integrated into a hydrogel for topical administration in the rat excision wound model. While the manuscript addresses an intriguing topic, notable revisions are required for it to meet the standards for publication.

Introduction:

The main objective of the study could be stated more explicitly at the beginning of the introduction. Clearly outlining the research gap or question that the study aims to address would help readers understand the significance of the research. The information is presented in a somewhat disjointed manner. Consider reorganizing the content to create a smoother flow of ideas, especially when transitioning between the general wound healing process, the use of phenolic compounds, and the introduction of gold nanoparticles.

Materials and methods:

I suggest that the authors offer a brief description of the extraction process for polyphenolic compounds from Bacopa procumbens, even if the procedure has been previously outlined. This additional description would serve to provide readers with a quick reference and enhance the overall clarity of the methodology.

To enhance the clarity of the experimental procedure, I suggest authors to provide detailed information on how the nanoparticles were added to the hydrogel and the specific blending method employed? Additionally, I would like to seek clarification on the quantity of gold nanoparticles that were introduced into the hydrogel.

Results

I recommend that the authors include SEM images of AuNP and AuNP-BP for a more comprehensive illustration.

It would be beneficial if the authors could furnish the average size of AuNP and AuNP-BP along with corresponding standard deviations to enhance the precision of the reported measurements.

The authors are encouraged to include a reference for the following section to support and provide additional context for the information presented.

“The average of skin wound healing with the treatments showed collagen bands, two intense bands at 1660 cm-1, and 1549 cm-1, are related to stretching vibration of C=O functional groups; a combination of C-N stretching and N-H bending vibration in the triple helix of collagens, 1338 cm-1 (CH2 side chain vibrations), 1286 cm-1 and 1204 cm-1 related to CH2 wagging vibration from the glycine backbone and proline sidechain and 1082 cm-1 , assigned to C-O stretching vibrations of the carbohydrate residue. The bands at 1737 cm-1 (C=O) and, 1456 cm-1 (CH2) are related with lipids, whereas the band at 1400 cm-1 arise from CH3 of GAGs; also phospholipids group absorbs at 1268 cm-1 and 1085 cm-1

I recommend that the authors maintain consistency in the format used to present peaks in different FTIR spectra. Whether utilizing numerical labels (1, 2, 3, ...) or including wavenumbers, maintaining a uniform presentation style across all spectra would enhance clarity and facilitate a smoother understanding for the readers.

I suggest the authors clarify the presentation of Tables 3 and 4 in Section “2.2.3. ATR-FTIR spectra changes on skin wound healing induced by AuNP-BP”. The current format is causing confusion. Additionally, despite the mention of significant shifts related to Amide I, Amide II, CH2 lipids, PO2- asym Phospholipids, and PO2- sym Phospholipids in Figure 8 and Table 3, it appears that no detectable significant differences between the spectra are apparent. Further clarification or reevaluation of the results may be needed to enhance the understanding of this section.

I would recommend the authors address the discrepancy between the mentioned shift on collagen bands (Amide I and Amide II at 1651 cm-1 and 1549 cm-1) to 1456 cm-1 and 1240 cm-1, respectively, in the discussion and the lack of visibility of such a shift in the actual FTIR spectrum. A clarification or potential reevaluation of the data could provide a more accurate representation of the findings.

I recommend authors enhance the conclusion by providing specific quantitative details, establishing a clearer cause-and-effect relationship, and offering more specific details about the observed macroscopic effects for enhanced clarity and precision.

Comments on the Quality of English Language

 Moderate editing of English language required

Author Response

In this investigation, aimed at evaluating the influence of AuNPs on enhancing the bioactivity of B. procumbens in tissue repair for skin injuries, and given the significance of biocompatibility and the extensive study of AuNPs, the authors undertook the synthesis and characterization of gold nanoparticles functionalized with polyphenolic compounds from B. procumbens (AuNPs-BP). Subsequently, the resulting colloidal dispersion was integrated into a hydrogel for topical administration in the rat excision wound model. While the manuscript addresses an intriguing topic, notable revisions are required for it to meet the standards for publication.

Introduction:

Comment. - The main objective of the study could be stated more explicitly at the beginning of the introduction. Clearly outlining the research gap or question that the study aims to address would help readers understand the significance of the research. The information is presented in a somewhat disjointed manner. Consider reorganizing the content to create a smoother flow of ideas, especially when transitioning between the general wound healing process, the use of phenolic compounds, and the introduction of gold nanoparticles.

Answer. – Thank you very much, you are right, thank you for your comment, we rewrote the introduction, we wrote sequentially the information, and we joined it properly for better understanding. (please see section from lines 53-89).

Materials and methods:

Comment. - I suggest that the authors offer a brief description of the extraction process for polyphenolic compounds from Bacopa procumbens, even if the procedure has been previously outlined. This additional description would serve to provide readers with a quick reference and enhance the overall clarity of the methodology.

Answer. – Thank you, we already added please see lines 404 - 409.

Comment. -To enhance the clarity of the experimental procedure, I suggest authors to provide detailed information on how the nanoparticles were added to the hydrogel and the specific blending method employed? Additionally, I would like to seek clarification on the quantity of gold nanoparticles that were introduced into the hydrogel.

Answer. – Thanks for your observation, we already include the following paragraph to explain the process: “AuNP-BP at 1.6 mg/ml all of them included into hydrogel. For hydrogel integration, in a reactor equipped with stirring blades, 70% deionized water was set under stirring at minimal velocity. To this, AuNP-BP, 1% glycerin, hydantoin, 0.2% methylchloroisothia-zolinone, and methylisothiazolinone were added gradually over 10 minutes. Then, 0.7% carbopol was introduced, with continuous stirring for an additional 10 minutes. Finally, 1% triethanolamine was added until gel formation was achieved. The mixing process was then maintained at medium velocity for 15 minutes.” Also, in the text we pointed out the amount of gold nanoparticles included. Please see lines 471-477).

Results

Comment. - I recommend that the authors include SEM images of AuNP and AuNP-BP for a more comprehensive illustration.

Answer. – Images of the gold nanoparticles (AuNPs), used for the elaboration of the BP conjugate, were obtained by transmission electron microscopy (TEM). These have been included in the manuscript for a better appreciation of their morphology. Please see figure 4 at section 2.1.3 (lines 155-167).

Comment. - It would be beneficial if the authors could furnish the average size of AuNP and AuNP-BP along with corresponding standard deviations to enhance the precision of the reported measurements.

Answer. –The average particle size of AuNPs, estimated by the TEM image processing with Image J software was 14.17±0.05 nm for AuNP, and 16.5±0.05 nm for AuNP-BP. The standard deviation was calculated by a Gaussian adjustment of the particle size distribution using the Origin software version 6.5. TEM imaging of AuNP-BP shows the nanoparticles are very close to each other, due to the presence of the BP extract on the surface of the particles. Thanks for the observation. Please see line 160.

Comment. - The authors are encouraged to include a reference for the following section to support and provide additional context for the information presented.

“The average of skin wound healing with the treatments showed collagen bands, two intense bands at 1660 cm-1, and 1549 cm-1, are related to stretching vibration of C=O functional groups; a combination of C-N stretching and N-H bending vibration in the triple helix of collagens, 1338 cm-1 (CH2 side chain vibrations), 1286 cm-1 and 1204 cm-1 related to CH2 wagging vibration from the glycine backbone and proline sidechain and 1082 cm-1 , assigned to C-O stretching vibrations of the carbohydrate residue. The bands at 1737 cm-1 (C=O) and, 1456 cm-1 (CH2) are related with lipids, whereas the band at 1400 cm-1 arise from CH3 of GAGs; also, phospholipids group absorbs at 1268 cm-1 and 1085 cm-1

Answer. – Thank you, we added the references related to the band’s assignment, on the main text, please see lines 236, 602-611.

Comment. - I recommend that the authors maintain consistency in the format used to present peaks in different FTIR spectra. Whether utilizing numerical labels (1, 2, 3, ...) or including wavenumbers, maintaining a uniform presentation style across all spectra would enhance clarity and facilitate a smoother understanding for the readers.

Answer. – Thank you, we modified the image to achieve uniformity, although in the image related to the bands that have significant differences, we kept the band assignment to facilitate the reading Figure 9 (line 238).

Comment. - I suggest the authors clarify the presentation of Tables 3 and 4 in Section “2.2.3. ATR-FTIR spectra changes on skin wound healing induced by AuNP-BP”. The current format is causing confusion. Additionally, despite the mention of significant shifts related to Amide I, Amide II, CH2 lipids, PO2- asym Phospholipids, and PO2- sym Phospholipids in Figure 8 and Table 3, it appears that no detectable significant differences between the spectra are apparent. Further clarification or reevaluation of the results may be needed to enhance the understanding of this section.

Answer. – Thaks for the observation, we added the missing information in the table, we also reevaluated the statistical analysis obtaining the same significant results, the changes might not be visible since it is only the median of FTIR, the analysis was performed using all the data considering the triplicates of each sample, considering that 6 wounds from each group were analyzed (1 wound for each animal). (Please see tables at lines 246 and 253)

Comment. - I would recommend the authors address the discrepancy between the mentioned shift on collagen bands (Amide I and Amide II at 1651 cm-1 and 1549 cm-1) to 1456 cm-1 and 1240 cm-1, respectively, in the discussion and the lack of visibility of such a shift in the actual FTIR spectrum. A clarification or potential reevaluation of the data could provide a more accurate representation of the findings.

Answer. –Thank you, we have a mistake on redaction, we rewrite the text and included in lines 373-377:

By FTIR spectroscopy, we found a shift on collagen bands amide I and amide II (1651 cm-1 and 1549 cm-1). These shifts suggest conformational changes on helices, which were analyzed by the increase in the ratio at 1657 cm-1-1652 cm-1/1689 cm-1-1679 cm-1, which showed an increase in AuNP-BP, demonstrating a better collagen alignment, as described by de Campos Vidal and Mello, 2019.

The results obtained by FTIR spectroscopy are consistent with Masson's trichrome and picrosirius red staining results, which showed better alignment and improvement on collagen deposition, similar findings related to collagen are observed by Chen et all 2023, as a result of treatment with nepebracteatalic acid nanoparticles (a diterpene) were improve collagen deposition.

Comment. - I recommend authors enhance the conclusion by providing specific quantitative details, establishing a clearer cause-and-effect relationship, and offering more specific details about the observed macroscopic effects for enhanced clarity and precision.

Answer. – Please see lines 524 – 532: The formulation and nanostructuring of polyphenolic compounds from B. procumbens with 16.5 nm AuNPs average size effectively promotes healing by reducing 100 times the concentration of BP required for tissue regeneration. The nanoformulation (AuNP-BP) applied topically accelerated wound healing, promoting 85.2% lesion reduction after 7 days. Histological changes showed a significant decrease in the inflammatory phase, increasing fibroblast proliferation and improving collagen organization. It induced collagen structural changes, evidenced by FTIR spectra, increasing the α-helix/β-sheet ratio from 1 in animals without treatment to 1.22 after topical treatment with the nanoformulation (AuNP-BP).

Round 2

Reviewer 2 Report

Comments and Suggestions for Authors

The manuscript has been significantly improved. It can be accepted as it is now. 

Author Response

Thank you for your review and approval

Reviewer 3 Report

Comments and Suggestions for Authors

I have no further comments. The revised manuscript merits publication in IJMS.

Author Response

Thank you for your review and approval

Reviewer 4 Report

Comments and Suggestions for Authors

Thanks to the authors for the updated manuscript! They've tackled most of my concerns, but I'm still a bit fuzzy on the details in Tables 3 and 4. Specifically, I'm wondering why the authors opted for presenting the wavenumber range (IQR). Also, the full name of IQR seems to be missing from the manuscript.

Comments on the Quality of English Language

 Minor editing of English language required

Author Response

29 November, 2023

Dear reviewer thanks for your previous comments to our manuscript, please you could find the response to your last comment to the revised version.

Comment. -Thanks to the authors for the updated manuscript! They've tackled most of my concerns, but I'm still a bit fuzzy on the details in Tables 3 and 4. Specifically, I'm wondering why the authors opted for presenting the wavenumber range (IQR). Also, the full name of IQR seems to be missing from the manuscript.

Thank you for your valuable’s observations

Answer. – Thank you for commenting your doubt, the main idea of including the IQR is because we want to show the dispersion of our data after the normalised analysis of the 18 measurements per group; considering that each group consisted of 6 animals, also the samples were measured in triplicate. The Shapiro-Wilk test was performed for each maximum peak and maximum absorbance, since all variables have p <0.05, then the Kruskal-Wallis test was applied.

Bellow you could find two reference in which this analysis were done:

- Vazquez-Zapien GJ, et al. Comparison of the Immune Response in Vaccinated People Positive and Negative to SARS-CoV-2 Employing FTIR Spectroscopy. Cells. 2022 1;11(23):3884. https://doi.org/10.3390/cells11233884

Martinez-Cuazitl A, et al. Biochemical, and ATR-FTIR Spectroscopic Parameters Associated with Death or Survival in Patients with Severe COVID-19". Journal of Spectroscopy. 2023. 2023:3423183. https://doi.org/10.1155/2023/3423183

For clarity, we have added the IQR meaning to the tables. Please see line 247 and 254.

Finally, the manuscript has been proofread by a native English speaker. In the manuscript, changes are highlighted in green.
